# Phenotypic Variability in Resistance to Anthracnose, White, Brown, and Blight Leaf Spot in Cassava Germplasm

**DOI:** 10.3390/plants13091187

**Published:** 2024-04-25

**Authors:** José Henrique Bernardino Nascimento, Luciano Rogerio Braatz de Andrade, Saulo Alves Santos de Oliveira, Eder Jorge de Oliveira

**Affiliations:** 1Centro de Ciências Agrárias, Ambientais e Biológicas, Universidade Federal do Recôncavo da Bahia, Cruz das Almas 44380-000, Bahia, Brazil; 2Embrapa Mandioca e Fruticultura, Cruz das Almas 44380-000, Bahia, Brazil; lucianorogerio@gmail.com (L.R.B.d.A.); saulo.oliveira@embrapa.br (S.A.S.d.O.)

**Keywords:** *Manihot esculenta* Crantz, diseases, selection, correlation, breeding

## Abstract

Despite fungal diseases affecting the aerial parts of cassava (*Manihot esculenta* Crantz) and causing significant yield losses, there is a lack of comprehensive studies assessing resistance in the species’ germplasm. This study aimed to evaluate the phenotypic diversity for resistance to anthracnose disease (CAD), blight leaf spot (BliLS), brown leaf spot (BLS), and white leaf spot (WLS) in cassava germplasm and to identify genotypes suitable for breeding purposes. A total of 837 genotypes were evaluated under field conditions across two production cycles (2021 and 2022). Artificial inoculations were carried out in the field, and data on yield and disease severity were collected using a standardized rating scale. The top 25 cassava genotypes were selected based on a selection index for disease resistance and agronomic traits. High environmental variability resulted in low heritabilities ($h^{2}$) for CAD, WLS, and BLS ($h^{2}$ = 0.42, 0.34, 0.29, respectively) and moderate heritability for BliLS ($h^{2}$ = 0.51). While the range of data for disease resistance was narrow, it was considerably wider for yield traits. Cluster analysis revealed that increased yield traits and disease severity were associated with higher scores of the first and second discriminant functions, respectively. Thus, most clusters comprised genotypes with hybrid characteristics for both traits. Overall, there was a strong correlation among aerial diseases, particularly between BLS and BliLS (r = 0.96), while the correlation between CAD and other diseases ranged from r = 0.53 to 0.58. Yield traits showed no significant correlations with disease resistance. Although the mean selection differential for disease resistance was modest (between −2.31% and −3.61%), selection based on yield traits showed promising results, particularly for fresh root yield (82%), dry root yield (39%), shoot yield (49%), and plant vigor (26%). This study contributes to enhancing genetic gains for resistance to major aerial part diseases and improving yield traits in cassava breeding programs.

## 1. Introduction

The *Manihot* genus belongs to the Euphorbiaceae family, which includes other economically significant species such as jatropha and rubber tree. These species diverged from cassava approximately 35 million years ago [1]. Cassava is globally recognized as one of the fastest-growing economically important crops due to its versatile applications across various industrial sectors [2]. Presently, Nigeria leads in cassava production, cultivating 60.0 million tons on 7.73 million hectares, while Brazil ranks fifth, producing 18.2 million tons across 1.21 million hectares [3]. Brazilian cassava production is estimated to be worth around USD 2.3 billion, supporting approximately one million direct and indirect jobs [4,5]. However, despite this economic significance, the predominant use of outdated varieties with low yields and high susceptibility to pests and diseases remains a challenge in various national production systems, leading to significant losses in root yield.

Among the primary diseases constraining cassava’s productive potential are those affecting the aerial parts of plants, including anthracnose complex (*Colletotrichum fructicola*, *C. tropicale*, *C. gloesoporiodes*, *Colletotrichum theobromicola*, and *C. siamense*), brown leaf spot (*Clarohilum henningsii*), white leaf spot (*Passalora manihotis*), and blight leaf spot (*Passalora vicosae*) [6,7]. These fungal diseases spread through water and wind, with higher incidences during rainy periods, which favor the growth and proliferation of these pathogens.

Anthracnose poses a threat to cassava cultivation throughout its developmental stages, causing deep cankers on stems, branches, and fruits, as well as leaf spots, wilting, stem breakage, and, in severe cases, plant death [8,9]. Yield losses due to anthracnose can reach staggering levels of up to 90% [10]. Conversely, brown leaf spot manifests as round lesions with a crescent shape, ranging from yellowish to brown, while blight leaf spot exhibits large, irregular spots without well-defined edges, often featuring large necrotic brown lesions [11,12,13]. White leaf spot displays symptoms of circular and angular lesions with a visible white color [14]. Studies suggest that foliar lesions lead to a significant reduction in leaf area and root yield, resulting in losses of up to 40% [15]. Typically, such losses can occur due to the redirection of starch from the roots towards the reconstitution of leaf area and aerial parts of the plants.

Management methods for these diseases in cassava cultivation include the use of resistant cultivars, crop rotation, and chemical control using fungicides [16]. However, chemical control of foliar lesions faces challenges, such as the limited availability of active ingredients registered in Brazil for their control [17].

Given that Brazil is the primary center of origin for cassava, its germplasm boasts wide genetic variability for attributes like root and starch yield and quality, adaptation to diverse soil and climatic conditions, and tolerance to pests and diseases [18,19,20]. Indeed, several authors have reported significant genotypic diversity in Brazilian cassava germplasm, supported by single-nucleotide polymorphism (SNP) markers, along with estimates suggesting an initial division of the original Amazon population into the Atlantic Forest and Caatinga ecoregions, with ongoing gene flows [21,22]. Therefore, this broad molecular variation may also reflect sufficient phenotypic diversity to identify sources of resistance to cassava foliar lesions.

Assessing accessions from germplasm banks is crucial for selecting potential parents to initiate breeding efforts [23,24]. While Embrapa Mandioca e Fruticultura have undertaken various actions to characterize and evaluate germplasm for attributes like flowering patterns, root yield and quality, starch quality, and resistance to root rot, consolidated efforts to evaluate a significant set of cassava clones for resistance to both the aerial parts (stems and foliage) remain limited. Hence, this study aims to estimate the phenotypic diversity in Brazilian cassava germplasm for resistance to anthracnose, as well as white (WLS), brown (BLS), and blight leaf spot (BliLS), and to correlate resistance patterns with yield attributes. The ultimate goal is to select genotypes with high resistance to aerial part diseases and suitable agronomic attributes, conducive to high-yield production standards.

## 2. Results

### 2.1. Diversity, Variances, and Genetic Parameters of Foliar Part Diseases and Yield Traits

The fixed effect of the evaluation for the different years was significantly different among cassava germplasm (*p* < 0.05) for all four aerial part diseases (Table 1). The same trend was observed in the deviance analysis of random effects of clone and genotype × year interaction (G × A) (Table 2). This demonstrates the presence of phenotypic diversity in cassava germplasm for all these aerial part diseases.

The values of genetic coefficient of variation ($CV_{g}$) were about two to three times lower than the coefficients of environmental variation ($CV_{e}$), namely, 6.95, 5.92, 5.59, and 18.16% of phenotypic variation (Table 3) for CAD, WLS, BLS, and BliLS, respectively. As a result of this high environmental effect on the phenotypic expression of aerial part diseases in cassava, broad-sense heritabilities ($h^{2}$) ranged from low (0.41, 0.34, and 0.29 for CAD, WLS, and BLS, respectively) to moderate magnitude (0.51 for BliLS) (Table 3).

The range of variation in BLUPs was relatively narrow for resistance to aerial part diseases, yet it exhibited significant diversity across agronomic traits in the cassava germplasm (Figure 1). For anthracnose, BLUPs ranged from 2.98 to 3.39, with a mean of 3.17. Although most of the germplasm scored between 3.0 and 3.5, similar patterns were observed for other traits associated with the foliar diseases, such as blight leaf spot (ranging from 3.18 to 3.56, mean of 3.33) and brown leaf spot (ranging from 3.17 to 3.55, mean of 3.34), with the highest density of data between scores of 3.0 and 3.5. Conversely, WLS had a mean of 1.45, with data density ranging from 1.24 to 1.98. As a result, the majority of evaluated genotypes exhibited moderate resistance to CAD, BLS, and BliLS, and greater resistance to WLS.

The phenotypic variation in cassava germplasm for agronomic traits was much broader than that observed for aerial part diseases resistance data. For instance, concerning plant vigor, scores ranged from ‘2’ to ‘4’, with a mean of 2.5, predominantly distributed between scores ‘2’ and ‘3’. Number of roots per plant varied from three to nine, with an average of 6.7 roots per plant, with most data falling between 5 and 8 roots per plant. The variation range of fresh shoot, fresh root, and dry root yield was also quite extensive. Fresh shoot yield ranged from 10 to 40 t·ha^−1^, with an average of 20.26 t·ha^−1^, primarily distributed between 12 and 28 t·ha^−1^, while for fresh root yield, the variation was from 10 to 40 t·ha^−1^, with an average of 18.17 t·ha^−1^, mainly concentrated between 10 and 25 t·ha^−1^. In the case of dry root yield, data ranged from 4 to 10 t·ha^−1^, with an average of 5.89 t·ha^−1^, concentrated between 4.5 and 7.0 t·ha^−1^. Leaf retention and dry matter content exhibited a narrower range of variation, with scores of 1.7 and 2.0 and averages of 1.83 and range of 30 and 38% and average of 35.14%, respectively. Most accessions showed leaf retention scores between 1.8 and 1.9, and dry matter content between 34 and 37%.

All cassava agronomic traits exhibited significant clone effects, except for leaf retention, where the only significant factor was the interaction of clone, location, and year (Appendix A). The variance attributed to clone was the most pronounced among all other factors, except for leaf retention, which did not differ significantly from zero (Appendix A). Location did not demonstrate significance for any of the agronomic traits, unlike year, which only showed significance for number of roots per plant (Appendix A). The triple interaction among clone, location, and year significantly impacted all agronomic traits, although it was not measured for plant vigor and number of roots per plant due to the absence of a location effect on those traits (Appendix A). Upon analyzing the interaction of clone with year, a significant effect was observed for dry root yield, dry matter content, fresh root yield, fresh shoot yield, plant vigor, and number of roots per plant, with leaf retention being the exception. The interaction between clone and location was significant only for fresh root yield and fresh shoot yield, yet it was more pronounced than the clone–year interaction for those traits. Similarly, the interaction between year and location had a significant impact comparable to the clone–year interaction, but only for dry root yield, fresh root yield, and fresh shoot yield (Appendix A).

### 2.2. Phenotypic Correlations

We conducted a discriminant analysis of principal components (DAPC) for the estimated traits using the K-means method. The first two discriminant functions explained 46.65% of the observed phenotypic variation (Figure 2). Upon reviewing the DAPC clustering, we observed no clear separation among cassava genotypes based on the first two discriminant functions, indicating some overlap among clusters in the two-dimensional plane. However, when considering the first three discriminant functions, we identified three distinct groups of cassava genotypes. These groups exhibited a relatively balanced distribution, with 345, 208, and 256 genotypes allocated to Groups ‘1’, ‘2’, and ‘3’, respectively. In general, an increase in the severity of aerial part diseases is associated with a higher score in the second discriminant function, while an increase in the values of yield traits correlates with an increase in the first discriminant function; thus, the DAPC clusters tend to encompass genotypes with characteristics related to both types of variables (disease resistance and yield potential).

In general, diseases of the aerial part showed high correlation among themselves, especially between brown leaf spot and blight leaf spot (r= 0.96) (Figure 3). On the other hand, the severity of white leaf spot showed moderate correlation with blight leaf spot (0.65) and brown leaf spot (0.64). Regarding anthracnose disease, the correlation was also of moderate magnitude (0.58 for BliLS and BLS, and 0.53 for WLS, respectively).

The yield traits dataset did not show any significant correlations with resistance to aerial part diseases of cassava (Figure 3). Therefore, considering only the correlation information between these two sets of characteristics, it is possible to identify and select cassava genotypes with greater susceptibility to leaf diseases, but with high yield potential and vice versa. On the other hand, there were correlations of high magnitude between yield traits such as fresh root yield with dry root yield (r= 0.79) and number of roots per plant (r= 0.50), as well as fresh shoot yield with fresh root yield (r= 0.58), dry root yield (r= 0.51), and plant vigor (r= 0.57). For the other pairs of agronomic traits, the correlations were only of low magnitude (ranging between 0.02 and 0.38).

### 2.3. Analysis of Phenotypic Diversity

The distribution and mean of resistance traits to aerial part diseases and yield attributes are depicted in Figure 4, taking into account the grouping identified by DAPC into three distinct groups. Genotypes from ‘Group 2’ exhibited a higher average severity of aerial part diseases compared to the other groups. Conversely, the average resistance of genotypes from Groups ‘1’ and ‘3’ was quite similar across all diseases, with the most resistant genotypes being predominantly found in these two groups.

In contrast, the higher susceptibility to diseases affecting the aerial parts observed in genotypes belonging to ‘Group 3’, however, exhibited higher averages for agronomic traits such as plant vigor, leaf retention, number of roots per plant, dry matter content, fresh shoot and root yield, as well as dry root yield (Figure 4). Conversely, genotypes from ‘Group 2’ displayed higher averages compared to ‘Group 1’ for traits like plant vigor, number of roots per plant, fresh shoot and root yield, and dry root yield. Genotypes belonging to ‘Group 3’ demonstrated greater resistance to diseases affecting the aerial parts and were also the most productive. However, although genotypes from ‘Group 1’ also exhibited high resistance to diseases affecting the aerial parts, their productive potential was lower than those from ‘Group 3’ for most traits. Genotypes from ‘Group 2’ displayed intermediate yield potential for traits such as vigor, number of roots per plant, fresh shoot and root yield, as well as dry root yield, despite being more susceptible to diseases affecting the aerial parts.

Regarding diseases resistance dataset, the Shannon–Weaver index was higher among cassava genotypes from ‘Group 1’ for blight leaf spot, ‘Group 2’ for anthracnose disease, and ‘Group 3’ for brown and white leaf spot (Table 4). Conversely, for agronomic traits, the Shannon–Weaver index was higher for vigor, number of roots per plant, leaf retention, and dry root yield in genotypes belonging to ‘Group 2’, and for fresh shoot and root yield traits in genotypes from ‘Group 3’. Genotypes from Groups ‘1’ and ‘3’ exhibited the highest values of phenotypic diversity, as indicated by the Shannon–Weaver index for dry matter content in the roots.

### 2.4. Genotype Selection for Recombination

In order to select the top 25 cassava genotypes for crossing and generation of new cassava progenies with higher resistance to aerial part diseases associated with high agronomic and productive attributes, a selection index with different weights for each characteristic was used. As a result, 18 germplasm accessions (BGM-0049, BGM-0058, BGM-0093, BGM-0120, BGM-0279, BGM-0287, BGM-0323, BGM-0389, BGM-0394, BGM-0442, BGM-0693, BGM-0714, BGM-0847, BGM-0901, BGM-1282, BGM-1626, BGM-2042, and BGM-2082) were selected, along with seven improved cultivars (BR-11-34-41, BR-11-34-45, BRS Amansa Burro, BRS Caipira, BRS Novo Horizonte, BRS Poti Branca, and BRS Tapioqueira) (Table 5).

Compared to the evaluated population, the average selection differential regarding resistance to aerial part diseases was quite low, considering the reduction in disease severity ranging from −2.31% for anthracnose to −3.61% for white leaf spot (Table 5). The reduction in severity of brown leaf spot and blight leaf spot was −2.52% and −2.66%, respectively. On the other hand, the selection differential for attributes related to vigor and growth was quite high, such as the potential increase in plant vigor by 26% and fresh shoot yield by 49%. For this set of characteristics, only leaf retention showed a low selection differential (1.2%) compared to the average of the evaluated population.

For productive attributes, the selection differential was quite high for fresh root yield (82%) and dry root yield (39%), as well as the average increase in number of roots per plant (22%). On the other hand, for attributes related to root quality, such as dry matter content, the average selection differential was quite reduced (0.7%). Nevertheless, it was found that overall, the selection of these 25 cassava genotypes for recombination and generation of new progenies has the potential to bring improvement to most of the evaluated characteristics, as indicated by the average increase in the selection index of 42% compared to all evaluated genotypes.

## 3. Discussion

### 3.1. Components of Variation for Resistance to Aerial Part Diseases in Cassava Germplasm

The variance of genotype × year interaction ($\sigma_{ga\;}^{2}$) was responsible for explaining a significant portion of the phenotypic variance of resistance to aerial part disease in cassava, followed by genetic variance ($\sigma_{g}^{2}$) and residual variance ($\sigma_{e}^{2}$). Phenotypic expressions of genotypes influenced by environmental conditions [25] can result in data heterogeneity and differential genotype performance in different environments [26]. Indeed, other authors have also reported a high environmental effect on anthracnose resistance in cassava genotypes evaluated in other regions [27]. In another study, Freitas et al. [20] evaluated the effects of inbreeding depression in S_1_ families for resistance to foliar diseases in cassava cultivation. These authors mentioned that the selection of clones resistant to leaf spots should consider evaluation in various locations/environments, considering the strong environmental effect on their expression, especially when evaluations are conducted under conditions of natural pathogen infection.

The higher coefficient of residual variation ($CV_{e}$) compared to genetic variation coefficient ($CV_{g}$) for all aerial part diseases reinforces the strong influence of environmental conditions, making the selection process more complex for these traits in genetic improvement programs. Results of this nature have been reported in other cassava pathosystems, such as the root rot complex caused by *Fusarium* spp. And *Phytophthora* spp., where $CV_{e}$ ranged from 55.28 to 93.46%, respectively, for these two pathogens [28].

Moderate magnitudes of heritability were found for anthracnose, white, and blight leaf spot resistance ($h^{2}$ = 0.42; 0.34 and 0.51, respectively), while low heritability was identified for brown leaf spot ($h^{2}$ = 29%). The heritabilities found in this study are quite similar to those reported by Dalarosa et al. [27] for resistance to bacterial blight and anthracnose ($h^{2}$ = 0.38 and 0.52, respectively), as well as those reported by Valentor et al. [29] in the analysis of cassava genotypes for resistance to cassava brown streak disease virus, with heritability values of $h^{2}$ = 0.64.

Given the significant impact of environmental factors on the genetic parameters associated with resistance to aerial part disease in cassava, there is a need to refine selection strategies to address these challenges effectively. One potential approach is to conduct genotype screening under controlled conditions to minimize environmental fluctuations during plant growth, inoculation, and assessment phases, thereby reducing the contributions of $\sigma_{ga\;}^{2}$ and $\sigma_{e}^{2}$. However, it is worth noting that currently, there is a lack of optimized mass screening platforms for anthracnose, brown, blight, and white leaf spot resistance in cassava under artificial conditions. Moreover, implementing such phenotyping methods would require specialized infrastructure, which may pose practical and financial constraints.

Another critical consideration is the importance of conducting phenotyping trials directly in field conditions to ensure the relevance of selection outcomes for target cultivation systems. Unlike controlled environments, field trials offer a more realistic assessment of trait expression, as demonstrated by previous studies showing discrepancies between traits measured in controlled settings and their performance in the field [30]. To mitigate environmental variability, it is essential to employ strategies such as precise soil moisture management through irrigation, selecting areas with consistent soil conditions (e.g., fertility, compaction, and organic matter content), uniform distribution of inoculants across experimental plots, and implementing effective and consistent management practices throughout the trial period. By optimizing these factors, it is possible to enhance heritability and ensure that selected traits align closely with field performance, thus facilitating more effective breeding efforts in cassava disease resistance.

### 3.2. Association of Resistance to Aerial Part Disease with Yield Attributes

Overall, there was high susceptibility to aerial part disease in the evaluated cassava germplasm during the two production cycles. Blight and brown leaf spot were the most aggressive diseases (with the highest average scores among the evaluated diseases) compared to others. The variation in BLUPs for brown leaf spot ranged from 3.13 to 3.55, while for blight leaf spot, this variation was from 3.08 to 3.56. On the other hand, the variation for white leaf spot resistance ranged from 1.24 to 1.99. Despite the high genetic variability present in the evaluated genotypes for numerous agronomic attributes, most of them showed only moderate resistance to aerial part diseases in cassava.

Compared to other cassava diseases that can render production unviable, such as cassava brown streak disease in Africa, where root yield losses can reach 100% [31], leaf spot diseases and cassava anthracnose have relatively lower potential for loss. Estimates of losses caused by leaf spot diseases can reach up to 40% [15], while losses from anthracnose can vary from 17 to 73%, depending on cultivar susceptibility [32]. Bacterial blight diseases in cassava have also been reported to cause significant reductions in fresh shoot and root yield, although a high and significant correlation between bacterial severity and agronomic performance of cassava genotypes has not been reported [33].

Despite lower losses compared to some African viruses or root rot diseases, aerial part diseases can render cassava farms unprofitable, especially when farmers rely on payment based on starch content. In such cases, aerial part diseases can drastically reduce starch content because they often cause extensive defoliation, especially under environmental conditions favorable for pathogens. In response, cassava plants mobilize stored starch in the roots to regenerate the aboveground parts, and if harvesting occurs during this period, the dry matter and starch content of the roots will be dramatically reduced at the entry of processing industries.

In light of this scenario, breeding programs have been working towards developing cultivars that exhibit more stable starch production throughout the year and in different environments by incorporating greater tolerance to these diseases. Nonetheless, the development and adoption of new cultivars remain one of the main challenges for cassava breeding programs [34], especially when there is limited variability for selection, as observed in this evaluated germplasm pool.

The correlation analysis reveals a distinct separation between correlated traits, with one group comprising resistance-associated traits and another comprising agronomic and yield traits. Notably, there were no significant correlations of high magnitude between these two groups. Although it is expected that the occurrence of aerial part diseases in cassava may impact yield attributes such as fresh and dry root yield, our data do not demonstrate significant correlations for these two traits. One possible explanation for this discrepancy could be that the inoculations were conducted under conditions conducive to pathogen dissemination in the target region, but nearly at the end of the crop cycle (twelve months after planting). At this developmental stage, plants already have their yield potential established; therefore, while the inoculations may have triggered the appearance and growth of pathogens, they likely did not result in substantial reductions in production. Future studies are underway to evaluate the potential yield losses from these diseases through earlier inoculations during the cassava growth cycle.

Correlations among aerial part diseases were notably high, particularly between brown and blight leaf spot, exhibiting a correlation coefficient of (*r* = 0.96). For other comparisons, correlations of moderate magnitude were observed (*r* ~ 0.64). Practically, phenotyping for brown and blight leaf spot resistance necessitates skilled personnel due to the challenges in distinguishing between their symptoms. Despite this difficulty, the lesions caused by both diseases exhibit striking similarities [11,35], resulting in a high correlation in disease severity.

In terms of yield data, the most substantial correlations were identified between fresh root yield and dry root yield (*r* = 0.79), as well as the number of roots per plant (*r* = 0.50). Additional correlations included fresh shoot yield × fresh root yield (*r* = 0.58), dry root yield (*r* = 0.51), and plant vigor (*r* = 0.57). Similar correlations between fresh and dry root yields were reported by Peprah et al. [36] in their evaluation of provitamin A-biofortified cassava varieties in Ghana. Another study by Sampaio Filho et al. [37] found a correlation of *r* = 0.70 between fresh root yield and fresh shoot yield. Hence, our correlation findings align with expectations for the analyzed characteristics.

Considering these insights, cassava leaf spot diseases (BLS, WLS, and BliLS) and anthracnose (CAD) may be classified as secondary diseases in certain regions. However, their occurrence remains crucial for ensuring business profitability, especially in scenarios where roots are commercialized for processing, entailing tight profit margins.

### 3.3. Clustering of Phenotypic Diversity for Aerial Part Disease Resistance and Yield Attributes

The discriminant analysis accounted for approximately 46.16% of the phenotypic variance, effectively revealing the dispersion and grouping of cassava genotypes into three distinct clusters based on leaf spot data, and yield and root quality information. These clusters exhibited significant differences from each other, with minimal association between leaf spot diseases and agronomic data. Genotypes within Group 2 were identified as the most susceptible to aerial part diseases, while those in Group 3 demonstrated, on average, superior agronomic performance regarding yield and root quality attributes.

The potential for clustering cassava genotypes based on disease resistance has been explored in previous studies. For example, Hohenfeld et al. [2] evaluated 148 genotypes for resistance to cassava root rot disease and assessed their yield potential under conditions of natural soil pathogen infestation. They grouped the genotypes into five clusters and demonstrated the ability of PCA to segregate extremely susceptible genotypes, characterized by low survival rates (1.97%) and higher disease index values (average ω = 95.11%). Conversely, one cluster comprised resistant genotypes with the highest average survival rate, lowest disease index, and highest averages for fresh shoot and root yield.

The effectiveness of the K-means method for forming groups with minimal variation has also been highlighted. Oliveira et al. [38] emphasized non-hierarchical clustering of cassava germplasm using qualitative data, encompassing yield, root quality traits, and the severity of anthracnose and bacterial blight diseases. In summary, clustering enables the segregation of genotypes based on common characteristics, facilitating future germplasm management, whether for conservation purposes or to guide the selection of genotypes for in-depth assessments of disease resistance, as well as comprehensive evaluations of yield potential, root quality, and starch content.

### 3.4. Genotype Selection for Population Improvement

Cassava breeding programs are aimed at integrating desirable agronomic traits to develop cultivars that exhibit resistance to pests and diseases, alongside high yield and root quality. Phenotypic characterization of cassava germplasm stands as a primary approach to identify genotypes with favorable traits for potential utilization in crop improvement programs.

The results of this study highlight that even under substantial pressure from aerial pathogens (with controlled artificial inoculations), the average fresh root yield in the germplasm remained notably high, approximately 18.17 t·ha^−1^ (ranging from 10 to 40 t·ha^−1^). Although this fresh root yield falls slightly below the potential of improved material (with an average of 23.8 t·ha^−1^ and a range of 14.8 t·ha^−1^ to 32.4 t·ha^−1^—Sampaio Filho et al. [37]), it is consistent with evaluations of germplasm in other studies, reporting an average of 16.49 t·ha^−1^ and a range between 5.41 and 33.62 t·ha^−1^ [39].

The dry matter content in the roots displayed a variation between 30% and 38% (with an average of 35.14%), falling within the expected range for a representative sample of cassava germplasm. This aligns with the variation of 8.4% to 45.4% (with an average of 34.14%) identified by Rabbi et al. [40] when evaluating 3232 genotypes from the International Institute of Tropical Agriculture (IITA) germplasm. Despite diseases directly affecting the aerial parts (resulting in leaf drop and branch drying), leading to a reduction in aboveground weight, the average fresh shoot yield stood at 20.26 t·ha^−1^ (ranging from 10 to 40 t·ha^−1^). These values surpassed those reported by Hohenfeld et al. [2], which ranged from 0.30 t·ha^−1^ to 27.03 t·ha^−1^ with an average of 13.67 t·ha^−1^, when evaluating 148 cassava genotypes in areas heavily infested with root rot-associated pathogens.

Considering the simultaneous need for population improvement across various traits, a selection index was applied to identify the top 25 clones for enhancing resistance to aerial part diseases in cassava, as well as for improving agronomic attributes related to growth, yield, and root quality. This selection revealed a modest differential selection for disease resistance (with reductions ranging from 2% to 4%), possibly due to the lower variation in BLUPs for these traits. These findings contrast with selection for resistance to other cassava diseases, such as root rot, where the use of selection indices enabled the selection of genotypes with reductions in root rot symptoms on the peel and pulp by 45.24% and 46.08%, respectively [28].

On the contrary, there was a significantly high selection differential, particularly for fresh root yield (82%), dry root yield (39%), and the number of roots per plant (22%). As a final result, the selection index differential of the top 25 clones resulted in a 42% higher selection index compared to all evaluated genotypes. In other cassava studies, the genetic gains obtained with the selection of the top 10% identified clones through the selection index were 15.13%, 0.39%, 14.38%, 24.95%, and 1.84% for fresh root yield, dry matter content, dry root yield, fresh shoot yield, and harvest index, respectively. This allowed for effective early selection in the first stage of clonal evaluation [41]. Therefore, the selection potential in the evaluated cassava germplasm in this study for agronomic traits is quite high and can effectively contribute to population improvement.

Selection indices are valuable tools in plant breeding as they enable the efficient selection of superior genotypes [42]. The use of selection indices in cassava cultivation for agronomic aspects such as yield, dry matter content, harvest index, and disease responses demonstrates the potential of using such tools. They allow for the identification of promising genotypes with high agronomic potential and resistance to pests and diseases [43].

The possibility of achieving genetic gains using different selection strategies is one of the main contributions of quantitative genetics. Selection methods facilitate effective guidance of the breeding process, prediction of outcomes, and decision-making based on scientific evidence of the efficiency of the breeding method. Therefore, although relatively limited success was achieved for resistance to aerial part diseases in cassava, the use of selection indices proved to be an effective strategy for obtaining simultaneous gains in various desirable traits.

## 4. Materials and Methods

### 4.1. Plant Material and Experiment Conduct

This study was conducted at Embrapa Mandioca e Fruticultura (Latitude: 12°40′12″ S, Longitude: 39°06′07″ W, Average Altitude: 220 m) and at the Federal University of Recôncavo da Bahia (Latitude 12°40′39″ S, Longitude 39°06′26″ W, Average Altitude: 226 m), both located in Cruz das Almas city (Bahia, Brazil). The region’s climate is hot and humid tropical, with no dry season, an average annual temperature of 24.5 °C, average annual relative humidity of 80%, and average annual rainfall of 1170 mm concentrated from March to August, followed by hotter months (September to February) (Figure 5A,B).

A total of 837 genotypes from the cassava germplasm bank (BAG) of Embrapa Mandioca e Fruticultura were assessed. The collection predominantly comprises local and improved cassava varieties originating from various regions of Brazil, with additional accessions from Colombia, Venezuela, Nigeria, and Panama. Among these accessions, 163 are classified as sweet cassava (<100 ppm of cyanogenic compounds) and 626 as bitter cassava (>100 ppm of cyanogenic compounds), while the characteristics of the remaining accessions have yet to be determined.

Field evaluations of the cassava germplasm were conducted over two production cycles in 2021 and 2022 at the Experimental Farm of the Federal University of Recôncavo da Bahia (UFRB). The experimental design employed was augmented block design (DBA), with 819 non-common accessions and 28 checks (local and commercial varieties) distributed across 15 blocks. Each block comprised approximately 144 non-common genotypes. Experimental plots consisted of two rows, each with 10 plants, spaced 0.90 m between rows and 0.80 m between plants, under rainfed conditions. Soil preparation involved plowing and two harrowings, followed by the opening of planting furrows approximately 15 cm deep. Planting was carried out manually during the rainy season in the Cruz das Almas region of Bahia, in May of both 2020 and 2021, using stakes measuring 16–18 cm in length. Planting followed recommended agricultural practices for the crop, including fertilization and pest control.

### 4.2. Collection of Diseased Leaves and Isolation of Pathogens

To prepare the inoculum, cassava leaves exhibiting characteristic symptoms of anthracnose, white, brown, and blight leaf spot were gathered from the germplasm bank. The isolation of fungi commenced by identifying and transporting the leaves to the Phytopathology Laboratory. Here, they underwent a thorough washing in running water, followed by the individualized cutting of fragments displaying disease symptoms. These fragments, featuring approximately 0.3 cm between the diseased tissue and the healthy part, underwent a 30 s immersion in 70% ethanol, followed by a 1 min treatment in a 5% sodium hypochlorite solution. Subsequently, they were rinsed thrice with sterile distilled water. Afterward, the fragments were laid out to dry on sterilized filter paper before being placed onto potato dextrose agar (PDA) medium. Incubation followed at 24 °C for 7 days under a 12 h photoperiod.

The fungal cultures were purified and underwent pathogenicity testing using the detached leaf method. For this, cassava variety leaves were distributed into transparent plastic boxes (gerboxes). Subsequently, the central region of each leaf lobe was punctured and inoculated with culture medium disks containing isolate structures. Controls, consisting of disks with PDA medium devoid of fungal growth, were employed alongside the treatment samples. All isolates induced characteristic lesions on standard cassava varieties (in vitro), thereby facilitating their complete isolation for subsequent field inoculations. The isolates considered pathogenic were identified by morphological means and based on the ITS region of the rDNA.

### 4.3. Preparation of Inoculum and Artificial Inoculations

To prevent the escape effect and standardize the disease incidence in the experiment, artificial inoculations were conducted via mechanical spraying throughout the entire experiment for subsequent evaluations using rating scales. To optimize symptom expression at the field level, we identified the most aggressive isolates through the pathogenicity test, estimating their virulence based on the lesioned area of inoculated leaves. Consequently, we utilized two isolates of *Colletotrichum* (*C. fructicola* and *C. tropicale*) and one isolate of other pathogens associated with aerial diseases for the inoculation process. For the preparation of each isolate’s inoculum, mycelium disks (2 mm in diameter) from colonies with 20 days of growth were placed in Erlenmeyer flasks containing 1.5 L of liquid Sabouraud dextrose agar medium (40 g/L dextrose, 10 g/L peptone, and 20 g/L agar with a pH of 5.6). The flasks were maintained under continuous agitation (110 rpm), at 25 °C, and a 12 h photoperiod for 15 days to allow for the growth and multiplication of pathogens. Subsequently, the colonized isolates were crushed using a 400 W blender.

To evaluate the presence of spores in the Sabouraud agar medium, 1 mL of the medium was extracted and diluted with 9 mL of distilled water. Subsequently, an approximate 1 mL aliquot was examined under a microscope for visualization of spores (conidia). Following confirmation of spore production, an aqueous suspension was prepared, and the conidia were counted using a Neubauer chamber. The suspension concentration was adjusted to 2.5 × 10^5^ spores per ml. For field inoculation, a Vulcan-type mechanical sprayer with a 600 L capacity was employed.

The evaluations were conducted during the most intense rainfall periods in the region (June 2021 and June 2022) (Figure 5A,B), at twelve months after planting (MAP), a time of increased disease incidence, primarily due to favorable climatic factors conducive to dissemination.

### 4.4. Evaluated Traits

The severity assessments caused by aerial part diseases in cassava were conducted according to the following rating scales [44]:(i)Anthracnose (CAD): (0) = no infection; (1) = small lesions on leaves and stems; (2) = few shallow cankers on the stem or leaves in the lower half of the plant; (3) = many stem cankers followed by distortion and/or leaf lesions in the upper half of the plant; (4) = many hard and woody lesions on the stem and/or leaves; (5) = highly severe attack with many lesions on hard and woody stems and severe necrosis in leaf axils followed by wilting and severe defoliation of the plant; and (6) = drying of all branches and/or plant death.(ii)White leaf spot (WLS): (0) = no symptoms; (1) = presence of some affected leaves in the lower third of the plant; (2) = >50% of leaves affected in the lower third of the plant; (3) = affected leaves in the middle and lower third; (4) = mild incidence distributed throughout the plant; (5) = moderate incidence distributed throughout the plant, along with yellowing and/or defoliation of the lower third; (6) = complete defoliation of the plant.(iii)Brown (BLS) and blight leaf spot (BliLS): (0) = no symptoms; (1) = presence of some affected leaves in the lower third of the plant; (2) = mild incidence in the lower third of the plant; (3) = moderate incidence in the middle and lower thirds, along with yellowing of affected leaves; (4) = severe incidence distributed throughout the plant, along with yellowing and/or defoliation; (5) = partial defoliation of the plant; (6) = complete defoliation of the plant.

In addition to resistance to aerial part disease, the following yield data were collected to correlate the effects of leaf spots on the following characteristics: (i) fresh root yield—FRY (t·ha^−1^); (ii) fresh shoot yield—FSY (t·ha^−1^); (iii) number of roots per plant—NRP; (iv) leaf retention using (LR) a rating scale, where 1 = less than 5% leaf retention, 2 = between 6 and 15% leaf retention, 3 = between 16 and 30% leaf retention, 4 = between 31 and 50% leaf retention, 5 = more than 50% retention; (v) plant vigor (Vigor): evaluated based on a rating scale, where 1 = low vigor, 3 = intermediate vigor, 5 = high vigor; (vi) dry matter content—DMC (%), obtained by weighing 3 to 5 kg of roots per plot using the method proposed by Kawano et al. [45]; and (vii) dry root yield—DRY (t·ha^−1^), obtained by multiplying FRY with DMC.

### 4.5. Data Analysis

Genetic parameters regarding resistance to aerial part diseases and the best linear unbiased predictions (BLUPs) of cassava accessions for resistance were estimated through mixed models analysis. The statistical model employed for effect estimation was $y_{ijkm}=b_{i}+c_{j}+a_{k}+i_{jk}+e_{ijkm}$, where $y_{ijkm}$ is the observation of the evaluated plant in plot ijkm; $b_{i}$ denotes the effect of block *I*, as a fixed effect $b_{i\;}\sim \;N\;\left(\mu_{bi\;};0\right)$; $c_{j}$ is the random effect of clone *j*, as random effect $c_{j}\sim \;N\;\left(0;\sigma_{c}^{2}\right)$; $a_{k}$ is the effect of year *k*, assumed as a random effect $a_{k}\sim \;N\;\left(0;\sigma_{a}^{2}\right)$; $i_{jk}$ is the random effect of genotype × year interaction $i_{jk}\sim \;N\;\left(0;\sigma_{i}^{2}\right);$ and $e_{ijkm}$ is the random effect of experimental error, $e_{ijkm}\;\sim \;N\;\left(0,\sigma^{2}\right).$ These analyses were conducted using the lme4 package version 1.1-14 [46]. BLUPs, variance components, and broad-sense heritability were estimated according to the following formula: $h^{2}=\frac{\sigma_{g}^{2}}{\sigma_{g}^{2}+\sigma_{ga\;}^{2}+\sigma_{e}^{2}}$, where $\sigma_{g}^{2}$ is the genetic variance, $\sigma_{ga\;}^{2}$ is the variance of genotype × year interaction, $\sigma_{e}^{2}$ indicates the residual variance. Additionally, phenotypic and genotypic correlations for the evaluated traits were estimated.

The estimated BLUPs were integrated with the overall corrected mean for the fixed effect of blocks and standardized for principal component analysis (PCA). PCA was performed to select the number of principal components with variance greater than one for use in discriminant analysis of principal components (DAPC).

DAPC, employing PCA for covariate reduction, the K-means method for individual grouping, and discriminant analysis for visualizing population structure [47], was utilized to estimate population structure. The K-means method groups clones by minimizing variance within groups, similar to conventional discriminant analysis. The number of groups was determined using the ‘find.clusters’ function of the adegenet package [48], which utilizes Bayesian information criterion (BIC) to identify the most likely number of clusters within the dataset.

The distributions of phenotypic data for each trait were analyzed using the Shannon–Weaver diversity index (H’), following the method proposed by Grenier et al. [49]. Initially, the number of observations obtained among six classes created based on the density of traits (range—rng; minimum—min; maximum—max) was estimated $\left(\leq min-\frac{rng}{6};\;\leq min-\frac{2\times rng}{6};\;\leq min-\frac{3\times rng}{6};\;\leq min-\frac{4\times rng}{6};\;\leq min-\frac{5\times rng}{6};\leq max\right)$. Subsequently, the observed frequency of each class was determined to calculate the Shannon–Weaver diversity index using the formula: $H^{\prime }=-\overset{n}{\underset{i=1}{\sum }}p_{i}\log_{e}p_{i}$, where $p_{i}$ is the observed frequency of class i, n is the number of phenotypic classes for a trait, $p_{i}$ is the relative frequency, and $log_{e}$ is the natural logarithm. All *H’* indices were normalized and divided by $log_{e}n$ to maintain values between 0 and 1 (from monomorphism to maximum phenotypic diversity). The total population diversity $H\prime_{t}$ was estimated, and the proportion of each group from DAPC was calculated. ${H^{\prime }}_{s}={H^{\prime }}_{r}/H\prime_{t}$, where $H\prime_{r}=1/n\sum H\prime_{t}$ is the mean phenotypic diversity within DAPC groups, and $G_{st}=\left(H\prime_{t}-H\prime_{r}\right)/H\prime_{t}$ represents the mean phenotypic diversity among DAPC groups.

The clustering information was used to assess variance and differences in clustering means using Scott-Knott means clustering (*p* < 0.05), employing a fixed model for all factors using R software version 4.1.0 [50]. The BLUPs for dry matter content, plant vigor, root number, leaf retention, dry root yield, fresh root, and fresh shoot yield were estimated for each accession using the following mixed model: $y_{ijl}=c_{i}+\beta_{j}+r_{l\left(j\right)}+e_{ijl}$, where $y_{ijkm}$ is the observation of plot ijl; $c_{i}$ is the effect of clone *i*, assumed as a random effect with $$c_{i}\sim N\left(0;\sigma_{c}^{2}\right)$; $\beta_{j}$ is the combination of location and year, assumed as a fixed effect with $\beta_{j}\sim N\left(\bar{\beta },0\right)$; $r_{l\left(j\right)}$ é the effect of repetition nested with the combination of location and year, assumed as a random effect with $r_{l\left(j\right)}\sim N\left(0,\sigma_{r}^{2}\right)$; and $e_{ijl}$ is the random effect of experimental error, assumed as a random effect with $e_{ijl}\sim N\left(0;\sigma^{2}\right)$. These analyses were performed using the lme4 package [46].

To select the top 25 cassava genotypes for disease resistance, root yield, and quality, a selection index based on the sum of ranks [51] and predefined weights was employed: $SI=\left(\;Anth\times -10\right)$ + $\left(BlLS\times -10\right)$ + BrLS×−10) + $\left(WhLS\times -5\right)$ + $\left(Vigor\times 5\right)$ + $\left(NRP\times 8\right)$ + $\left(LR\times 10\right)$ + $\left(DRY\times 20\right)$ + $\left(DMC\times 10\right)$ + $\left(FRY\times 10\right)$ + FSY×5), where IS refers to the sum of BLUPs for each trait multiplied by their respective weights.

## 5. Conclusions

The phenotypic variability observed within the cassava germplasm was notably high, particularly in yield traits, albeit somewhat constrained concerning resistance to aerial diseases. Nonetheless, selecting the top 25 cassava genotypes for recombination aimed at population improvement yielded a significantly robust selection differential (>22%) for the selection index and key yield traits like fresh root yield, dry root yield, and number of roots per plant. Although only modest reductions in disease severity (between 2 and 4%) were observed.

These findings suggest that despite some susceptibility to aerial diseases, it remains feasible to identify and select cassava genotypes with high yield potential. However, it is crucial that genotype selection also prioritizes resistance, especially considering the adverse impact certain diseases like anthracnose can have on propagative material quality.

The genotypes within the assessed cassava germplasm showcase traits of considerable interest for breeding programs. Their selection promises to bolster genetic gains and facilitate the development of cultivars better suited to diverse cassava production systems.

## Figures and Tables

**Figure 1 plants-13-01187-f001:**
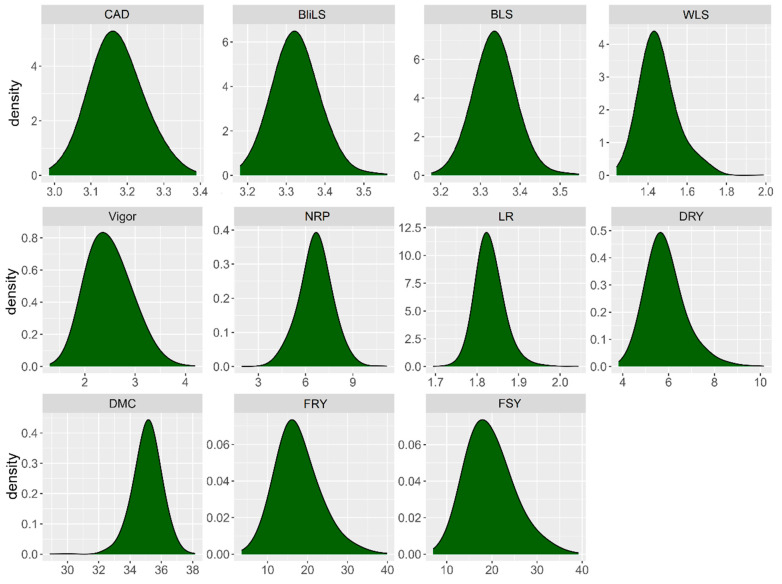
Distribution of the best linear unbiased predictors (BLUPs) for aerial part diseases such as cassava anthracnose disease (CAD), white leaf spot (WLS), brown leaf spot (BLS), and blight leaf spot (BliLS), along with agronomic traits including number of roots per plant (NRP), dry root yield (DRY), dry matter content (DMC), fresh root yield (FRY), fresh shoot yield (FSY), leaf retention (LR), and plant vigor (Vigor).

**Figure 2 plants-13-01187-f002:**
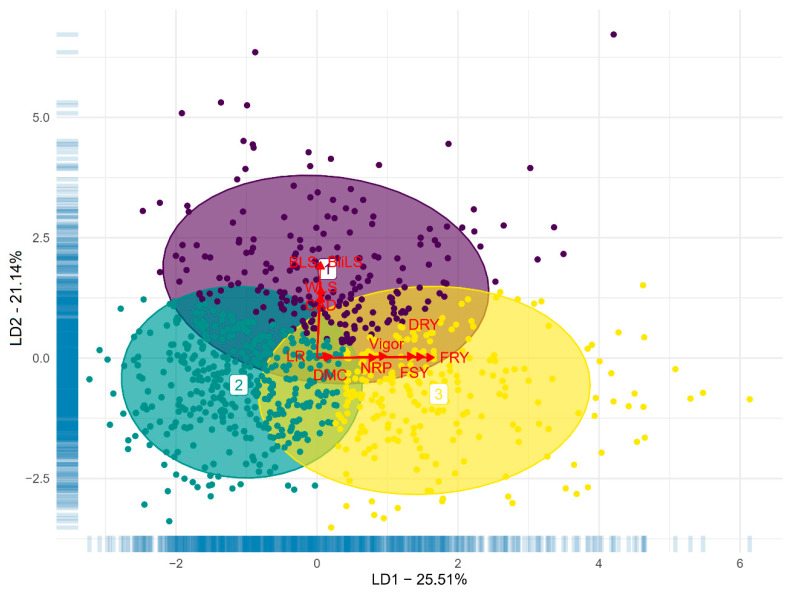
Scatter plot of the first and second discriminant function of the discriminant analysis of principal components based on evaluations of resistance to cassava anthracnose disease (CAD), white leaf spot (WLS), brown leaf spot (BLS), and blight leaf spot (BliLS), and agronomic variables such as number of roots per plant (NRP), dry root yield (DRY), dry matter content (DMC), fresh root yield (FRY), fresh shoot yield (FSY), leaf retention (LR), and plant vigor (Vigor). Numbers 1 (purple), 2 (green), and 3 (yellow) correspond to ‘Group 1’, ‘Group 2’, and ‘Group 3’, respectively.

**Figure 3 plants-13-01187-f003:**
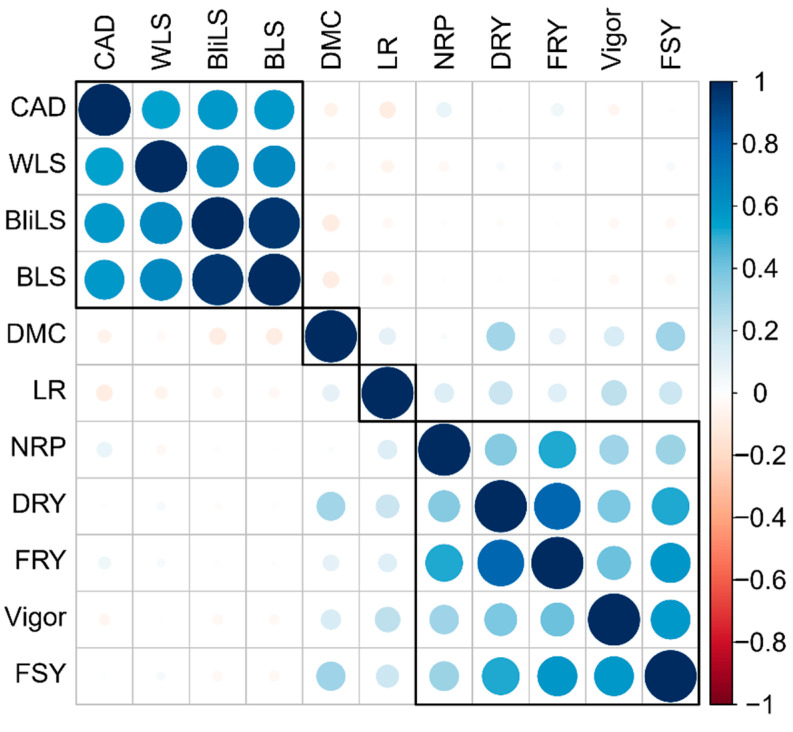
Correlogram between diseases of the aerial part and agronomic traits of cassava genotypes evaluated during the crop years of 2021 and 2022. Cassava anthracnose disease (CAD); white leaf spot (WLS); blight leaf spot (BliLS); brown leaf spot (BLS); dry matter content (DMC); leaf retention (LR); number of roots per plant (NRP); dry root yield (DRY); fresh root yield (FRY); plant vigor (Vigor); and fresh shoot yield (FSY).

**Figure 4 plants-13-01187-f004:**
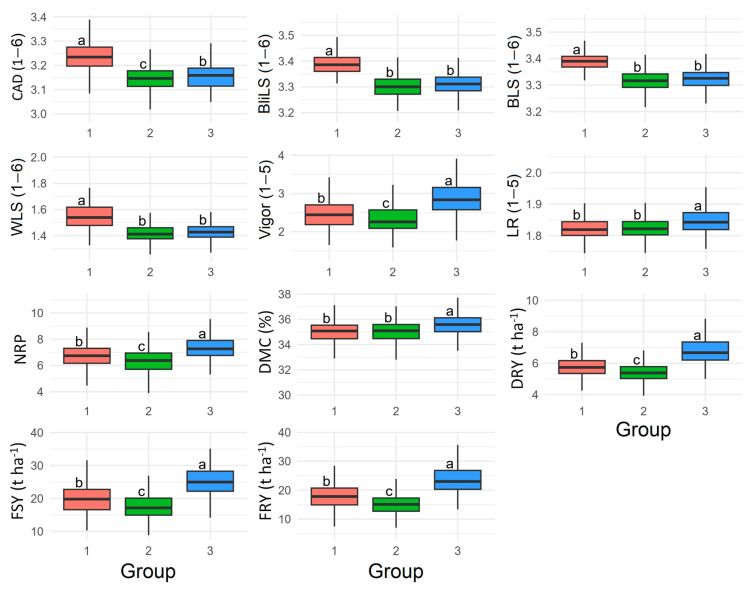
Boxplots of the best unbiased linear predictors of cassava genotypes grouped by principal component discriminant analysis regarding resistance to cassava anthracnose disease (CAD), white leaf spot (WLS), brown leaf spot (BLS), and blight leaf spot (BliLS), as well as agronomic and root quality traits such as plant vigor (Vigor), leaf retention (LR), number of roots per plant (NRP), dry matter content (DMC), dry root yield (DRY), fresh shoot yield (FSY), and fresh root yield (FRY). Numbers 1 (red), 2 (green), and 3 (blue) correspond to ‘Group 1’, ‘Group 2’, and ‘Group 3’, respectively. Different letters indicate significant differences.

**Figure 5 plants-13-01187-f005:**
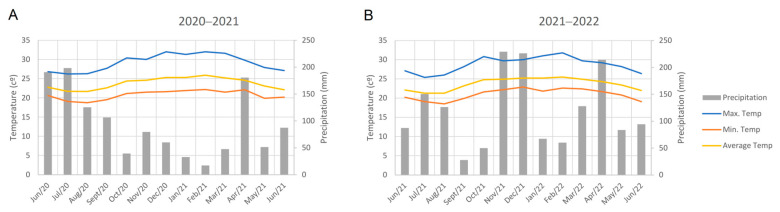
Meteorological data from the experimental area in the two production cycles: (**A**) 2020–2021 and (**B**) 2021–2022. Average Temp. = average temperature; Max. Temp. = maximum temperature; Min. Temp. = minimum temperature; Precipitation = accumulated precipitation.

**Table 1 plants-13-01187-t001:** Analysis of variance of fixed effects for the assessment of aerial part diseases in cassava accessions during the years 2021 and 2022, at 12 months after planting.

Disease	Factor	DF	SS	MS	F
CAD	Year	1	14.23	14.23	48.88 *
	Block	28	308.68	11.02	33.22 ^ns^
WLS	Year	1	111.25	111.25	276.94 *
	Block	28	264.52	9.45	23.52 ^ns^
BLS	Year	1	76.57	76.57	179.61 *
	Block	28	336.92	12.00	28.23 ^ns^
BliLS	Year	1	310.44	310.44	590.33 *
	Block	28	269.55	9.56	18.17 ^ns^

CAD = cassava anthracnose disease; WLS = white leaf spot; BLS = brown leaf spot; BliLS = blight leaf spot; DF: degrees of freedom; SS: sum of squares; MS: mean square; * significant at 5% probability by F test; ns: not significant.

**Table 2 plants-13-01187-t002:** Deviance analysis of random effects of clone and genotype × year interaction (G × A) for aerial part diseases during two production cycles at 12 months after planting.

Disease	Factor	Deviance	DF	*p*-Valor
CAD	Genotype (G)	1165.85	1	<0.01 **
	G × Year	2185.42	1	
WLS	Genotype (G)	865.05	1	<0.01 **
	G × Year	1298.36	1	
BLS	Genotype (G)	807.06	1	<0.01 **
	G × Year	1521.75	1	
BliLS	Genotype (G)	835.09	1	<0.01 **
	G × Year	1550.09	1	

CAD = cassava anthracnose disease; WLS = white leaf spot; BLS = brown leaf spot; BliLS = blight leaf spot; DF: degrees of freedom; **: significant at 5% probability by chi-square (χ^2^) test.

**Table 3 plants-13-01187-t003:** Genetic parameters of resistance to aerial part diseases in cassava at 12 months after planting during two production cycles (2021–2022).

Disease	$\sigma_{g}^{2}$	$\;\sigma_{ga\;}^{2}$	$\sigma_{e}^{2}$	$h^{2}$	$CV_{g}$(%)	$CV_{e}$(%)
CAD	0.49	0.80	0.33	0.41	6.95	18.17
WLS	0.39	0.68	0.40	0.34	5.92	19.06
BLS	0.34	0.71	0.43	0.29	5.59	19.57
BliLS	0.70	0.77	0.53	0.51	18.16	49.84

CAD = cassava anthracnose disease; WLS = white leaf spot; BLS = brown leaf spot; $\mathrm{BliLS}\;=\;\mathrm{blight}\;\mathrm{leaf}\;\mathrm{spot};\;\sigma_{g}^{2}$: genetic variance, variance of genotype × year interaction; $\sigma_{e}^{2}$: residual variance; $h^{2}$: broad-sense heritability; *CV_g_*: coefficient of genetic variation, *CV_e_*: coefficient of environmental variation.

**Table 4 plants-13-01187-t004:** Shannon–Weaver diversity index based on aerial part disease resistance and agronomic attributes per group formed by principal component discriminant analysis.

Group	CAD	BliLS	BLS	WLS	Vigor	NRP	LR	DRY	DMC	FRY	FSY
1	0.79	0.82	0.76	0.80	0.74	0.86	0.71	0.79	0.79	0.78	0.78
2	0.84	0.77	0.75	0.70	0.89	0.88	0.83	0.85	0.67	0.75	0.85
3	0.79	0.79	0.79	0.82	0.88	0.73	0.69	0.76	0.77	0.87	0.86
Average	0.81	0.79	0.76	0.77	0.84	0.82	0.74	0.80	0.74	0.80	0.83

Cassava anthracnose disease (CAD); blight leaf spot (BliLS); brown leaf spot (BLS); white leaf spot (WLS); plant vigor (Vigor); number of roots per plant (NRP); leaf retention (LR); dry root yield (DRY); dry matter content (DMC); fresh root yield (FRY); and fresh shoot yield (FSY).

**Table 5 plants-13-01187-t005:** Selection of the top 25 cassava genotypes for crossing aiming to increase disease resistance associated with yield and root quality attributes based on the selection index.

Genotype	CAD	BliLS	BLS	WLS	Vigor	NRP	LR	DRY	DMC	FRY	FSY	SelInd
BR-11-34-41	3.18	3.32	3.31	1.33	3.85	11.16	1.94	8.83	32.15	38.41	32.35	719.03
BRS Caipira	3.07	3.17	3.34	1.39	3.58	10.82	1.82	8.44	35.66	35.32	32.30	718.32
BGM-0442	3.10	3.20	3.21	1.60	3.01	8.44	1.84	7.78	34.87	37.69	33.24	706.39
BGM-0847	3.03	3.24	3.23	1.32	3.25	7.72	1.82	9.19	35.95	34.02	27.83	691.10
BGM-2082	3.11	3.21	3.17	1.38	3.61	8.30	1.88	7.41	35.67	35.30	25.48	690.48
BRS Novo Horizonte	3.04	3.23	3.27	1.42	3.50	9.23	1.89	8.50	36.56	32.68	34.04	689.79
BGM-0049	2.99	3.26	3.24	1.36	3.22	6.94	1.85	8.58	36.83	33.56	38.69	683.06
BGM-0093	3.21	3.34	3.35	1.51	2.79	7.49	1.85	8.78	35.27	35.33	32.56	680.62
BGM-2042	3.03	3.27	3.28	1.35	3.66	8.46	1.86	7.70	36.55	33.18	31.26	680.42
BRS Tapioqueira	3.10	3.24	3.36	1.47	3.32	9.85	1.90	7.78	35.40	33.08	27.65	675.46
BGM-0287	3.14	3.26	3.23	1.28	2.79	7.52	1.83	7.34	35.05	35.63	28.77	673.81
BRS Poti Branca	3.01	3.22	3.30	1.37	3.58	9.41	1.84	7.85	35.28	32.48	33.97	670.47
BGM-1626	3.06	3.19	3.21	1.31	2.73	7.97	1.82	7.95	35.25	33.80	28.19	670.04
BRS Poti-Branca	3.00	3.14	3.32	1.46	3.05	9.54	1.94	8.69	35.84	30.92	31.63	668.79
BGM-0901	3.08	3.08	3.18	1.33	3.33	8.52	1.80	8.37	35.64	31.76	30.78	667.14
BR-11-34-45	3.12	3.32	3.20	1.40	3.91	8.24	1.92	9.23	35.55	31.32	26.71	665.77
BGM-0279	3.15	3.17	3.22	1.50	3.26	8.18	1.81	8.57	34.86	32.51	29.98	660.80
BGM-0058	3.07	3.19	3.17	1.36	3.37	8.32	1.83	7.31	34.12	33.26	30.68	653.24
BGM-1282	3.02	3.19	3.13	1.32	3.02	6.35	1.81	9.15	36.85	30.16	34.83	652.11
BGM-0394	3.05	3.37	3.21	1.49	2.83	7.75	1.83	7.82	35.15	32.76	27.89	651.76
BGM-0693	3.09	3.22	3.22	1.35	2.78	7.79	1.83	7.86	35.30	32.25	26.48	651.19
BGM-0120	3.16	3.25	3.24	1.37	2.79	7.52	1.90	8.52	33.66	33.14	33.58	646.37
BGM-0714	3.05	3.22	3.25	1.26	3.03	8.06	1.87	7.21	35.51	31.76	29.38	646.22
BRS Amansa Burro	3.16	3.18	3.19	1.40	3.30	8.64	1.88	8.42	36.94	28.53	28.66	645.61
BGM-0323	3.15	3.17	3.28	1.37	2.90	7.76	1.83	8.28	35.98	30.43	23.71	642.82
Média selecionados	3.09	3.23	3.24	1.39	3.22	8.40	1.86	8.22	35.43	33.17	30.43	672.03
Média população	3.17	3.32	3.33	1.45	2.50	6.71	1.83	5.90	35.15	18.22	20.31	471.90
Diferencial de seleção (%)	−2.59	−2.90	−2.68	−4.55	28.49	25.20	1.29	39.47	0.81	82.06	49.82	42.41

Cassava anthracnose disease (CAD); blight leaf spot (BliLS); brown leaf spot (BLS); white leaf spot (WLS); plant vigor (Vigor); number of roots per plant (NRP); leaf retention (LR); dry root yield (DRY); dry matter content (DMC); fresh root yield (FRY); fresh shoot yield (FSY); and selection index (SelInd).

## Data Availability

All datasets generated for this study can be found in the article and Figshare (https://doi.org/10.6084/m9.figshare.25368313).

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
