# Peer review of "Phenotypic Variability in Resistance to Anthracnose, White, Brown, and Blight Leaf Spot in Cassava Germplasm"

_plants, 2024, doi:10.3390/plants13091187_

Round 1
Reviewer 1 Report
Comments and Suggestions for Authors
This is undoubtedly very valuable work. The authors tested a large amount of Cassava plant material for susceptibility to four diseases ie. anthracnose (CAD), blight leaf spot (BliLS), brown leaf spot (BLS), and white leaf spot (WLS). Set of informative parameters for each of 837 genotypes were assessed and these huge number of collected results were subjected to proper statistical analysis. The results were presented in a coherent, clear and communicative manner. However, I have a few questions/comments
The authors state that the cause of Cassava anthracnose is Colletotrichum spp. Because the species responsible for this disease have been identified, perhaps it is worth mentioning them in the introduction, even if the species was not identified for the purpose of inoculation tests.
The authors provide statistics for genotypes, genotypes x year, maybe it is worth showing statistics for interactions: genotype x environment, and genotype x environment x year. In this case environment means locality.
It is not clear how many isolates of particular fungal species were used for plant inoculation
It is not clear if the same plants were inoculated together with the four species of fungi or four sets of plants were inoculated with plant pathogen separately.
Authors should explain or add the comments why generally plant infection did not corelate with yields parameters (DRY and FRY) although intuitively such a relationship was to be expected
Author Response
The authors state that the cause of Cassava anthracnose is Colletotrichum spp. Because the species responsible for this disease have been identified, perhaps it is worth mentioning them in the introduction, even if the species was not identified for the purpose of inoculation tests.
Response: We state Cassava anthracnose is caused by Colletotrichum spp. based on our previous works with the disease (de Oliveira, Saulo Alves Santos, et al. "Colletotrichum species causing cassava (Manihot esculenta Crantz) anthracnose in different eco-zones within the Recôncavo Region of Bahia, Brazil." Journal of Plant Diseases and Protection 127 (2020): 411-416.; Bragança, C. A. D., et al. "First report of Colletotrichum fructicola causing anthracnose in cassava (Manihot esculenta Crantz) in Brazil." Plant Disease 100 (2016): 857.. Oliveira, S. A. S., et al. "First report of Colletotrichum theobromicola and C. siamense causing anthracnose on cultivated and wild cassava species in Brazil." Plant disease 102.4 (2018): 819-819.).
The information regarding the species was incorporated into the text
The authors provide statistics for genotypes, genotypes x year, maybe it is worth showing statistics for interactions: genotype x environment, and genotype x environment x year. In this case environment means locality.
Response: For the evaluation of cassava diseases, the study was limited to two years of assessment at a single location. Nonetheless, we will include a supplementary table detailing all the interactions among the various agronomic attributes of cassava, as requested by the reviewer.
It is not clear how many isolates of particular fungal species were used for plant inoculation
Response: Two isolates were used, being one Colletotrichum fructicola and other C. tropicale, and one isolate of pathogens associated with the remains diseases.
The information has been included in the text.
It is not clear if the same plants were inoculated together with the four species of fungi or four sets of plants were inoculated with plant pathogen separately.
Response: We added the following information to the text
“To optimize symptom expression at the field level, we identified the most aggressive isolates through the pathogenicity test, estimating their virulence based on the lesioned area of inoculated leaves. Consequently, we utilized two isolates of Colletotrichum (C. fructicola and C. tropicale) and one isolate of other pathogens associated with aerial diseases for the inoculation process.”
Authors should explain or add the comments why generally plant infection did not correlate with yields parameters (DRY and FRY) although intuitively such a relationship was to be expected
Response: We added the following explanation.
“Although it is expected that the occurrence of aerial part diseases in cassava may impact yield attributes such as fresh and dry root yield, our data do not demonstrate significant correlations for these two traits. One possible explanation for this discrepancy could be that the inoculations were conducted under conditions conducive to pathogen dissemination in the target region, but nearly at the end of the crop cycle (twelve months after planting). At this developmental stage, plants already have their yield potential established; therefore, while the inoculations may have triggered the appearance and growth of pathogens, they likely did not result in substantial reductions in production. Future studies are underway to evaluate the potential yield losses from these diseases through earlier inoculations during the cassava growth cycle.”
Reviewer 2 Report
Comments and Suggestions for Authors
The manuscript describes the evaluation of 837 cassava genotypes on their resistance against several aerial fungal diseases (anthracnose, white, brown, and blight leaf spot) and their association with yield attributes.
The amount of the described work is substantial, and its impact is considered significant since it provides selected cassava genotypes (n=25) to initiate breeding efforts.
More specifically, the performed selection process revealed a modest differential selection for disease resistance (with reductions ranging from 2% to 4%), while on the contrary, there was a significantly high selection differential, particularly for root yield, dry root yield, and the number of roots per plant. Therefore, the selection potential in the evaluated cassava germplasm in this study mainly for agronomic traits was considered quite high and can effectively contribute to population improvement.
The manuscript is very well written, the experimental procedure is properly organized to answer the scientific questions set by the authors, and overall, the research outcomes are very promising and well presented.
Comments on the Quality of English LanguageThe quality of English language is satisfying.
Author Response
The manuscript describes the evaluation of 837 cassava genotypes on their resistance against several aerial fungal diseases (anthracnose, white, brown, and blight leaf spot) and their association with yield attributes.
The amount of the described work is substantial, and its impact is considered significant since it provides selected cassava genotypes (n=25) to initiate breeding efforts.
More specifically, the performed selection process revealed a modest differential selection for disease resistance (with reductions ranging from 2% to 4%), while on the contrary, there was a significantly high selection differential, particularly for root yield, dry root yield, and the number of roots per plant. Therefore, the selection potential in the evaluated cassava germplasm in this study mainly for agronomic traits was considered quite high and can effectively contribute to population improvement.
The manuscript is very well written, the experimental procedure is properly organized to answer the scientific questions set by the authors, and overall, the research outcomes are very promising and well presented.
Response: We appreciate your thorough review of our manuscript and the complimentary comments on our work.
Reviewer 3 Report
Comments and Suggestions for Authors
Dear authors
This study highlights the significant phenotypic variability existing in Brazilian cassava germplasm. The mathematical methods applied along with experimental design showed that the selection of appropriate genotypes can result in robust improvements in key yield traits although disease severity reductions appeared to be modest. Genotype selection towards plant resistance is crucial considering the adverse impact of these diseases in the field.
The results achieved offer promise for breeding programs, enhancing genetic gains, and developing cultivars suited to diverse cassava production systems.
I consider the article suitable for publication, with adequate discussion and conclusions.
Just minor comments:
Line 87- In these sentence is best to enunciate what parameters were used in the evaluations performed. Did you consider Foliar Part Diseases and Yield Traits or only Foliar part diseases?
Figure 2: What do the 1, 2, and 3 correspond to in the figure? This is not explained in the legend. The legend must be self explanatory.
Line 484: How did you complete the isolation? Did you obtain isolates in pure culture? Or did you only perform pathogenicity tests? Did you perform other kinds of tests to correctly identify the causal agents of these diseases, such as molecular markers? The inoculation was performed with a mixture of isolates? Can you ensure that each inoculation matches exactly to the disease studied?
Author Response
Line 87- In these sentence is best to enunciate what parameters were used in the evaluations performed. Did you consider Foliar Part Diseases and Yield Traits or only Foliar part diseases?
Response: Both, foliar disease (Tables 1 to 3) and yield traits (Figure 1)
Figure 2: What do the 1, 2, and 3 correspond to in the figure? This is not explained in the legend. The legend must be self explanatory.
Response: An explanation has been added to the caption
Line 484: How did you complete the isolation? Did you obtain isolates in pure culture? Or did you only perform pathogenicity tests?
Response: We obtained pure cultures and conducted pathogenicity tests with the isolates. The corresponding information has been included in the text
Did you perform other kinds of tests to correctly identify the causal agents of these diseases, such as molecular markers?
Response: The isolates identified as pathogenic were characterized morphologically and through analysis of the ITS region of the rDNA. This information has been incorporated into the text.
The inoculation was performed with a mixture of isolates? Can you ensure that each inoculation matches exactly to the disease studied?
Response: Following separate cultivation, the isolates were mixed after concentration adjustments prior to inoculation. This additional detail has been included in the text.